# Nosological and Theranostic Approach to Vascular Malformation through cfDNA NGS Liquid Biopsy

**DOI:** 10.3390/jcm11133740

**Published:** 2022-06-28

**Authors:** Viola Bianca Serio, Maria Palmieri, Lorenzo Loberti, Stefania Granata, Chiara Fallerini, Massimo Vaghi, Alessandra Renieri, Anna Maria Pinto

**Affiliations:** 1Medical Genetics Unit, University of Siena, Policlinico “Santa Maria alle Scotte”, Viale Bracci, 2-53100 Siena, Italy; viola.serio@dbm.unisi.it (V.B.S.); maria.palmieri@dbm.unisi.it (M.P.); lorenzo.loberti@dbm.unisi.it (L.L.); stefania.granata@dbm.unisi.it (S.G.); fallerini2@unisi.it (C.F.); 2Med Biotech Hub and Competence Center, Department of Medical Biotechnologies, University of Siena, 2-53100 Siena, Italy; 3Genetica Medica, Azienda Ospedaliera Universitaria Senese, 2-53100 Siena, Italy; annamaria.pinto@dbm.unisi.it; 4Radiologia Interventistica, Ospedale Maggiore di Crema, Largo Ugo Dossena, 2-26013 Crema, Italy; vaghim@yahoo.it; 5Chirurgia Vascolare, Ospedale Maggiore di Crema, Largo Ugo Dossena, 2-26013 Crema, Italy

**Keywords:** NGS-liquid biopsy, arteriovenous malformations, lymphovenous malformations, venous malformations, tailored therapy

## Abstract

Several different nosological classifications have been used over time for vascular malformations (VMs) since clinical and pathological signs are largely overlapping. In a large proportion of cases, VMs are generated by somatic mosaicism in key genes, belonging to a few different molecular pathways. Therefore, molecular characterization may help in the understanding of the biological mechanisms related to the development of pathology. Tissue biopsy is not routinely included in the diagnostic path because of the need for fresh tissue specimens and the risk of bleeding. Bypassing the need for bioptic samples, we took advantage of the possibility of isolating cell-free DNA likely released by the affected tissues, to molecularly characterize 53 patients by cfDNA-NGS liquid biopsy. We found a good match between the identified variant and the clinical presentation. *PIK3CA* variants were found in 67% of Klippel Trenaunay Syndrome individuals; *KRAS* variants in 60% of arteriovenous malformations; *MET* was mutated in 75% of lymphovenous malformations. Our results demonstrate the power of cfDNA-NGS liquid biopsy in VMs clinical classification, diagnosis, and treatment. Indeed, tailored repurposing of pre-existing cancer drugs, such as PIK3CA, KRAS, and MET inhibitors, can be envisaged as adjuvant treatment, in addition to surgery and/or endovascular treatment, in the above-defined VMs categories, respectively.

## 1. Introduction

The improvements in the knowledge of biological processes are linked to the technological skills in diagnostics and lead to therapeutics development. This is especially true for vascular malformations that display a wide clinical variability and an important genetic heterogeneity with a consistent clinical overlap.

Vascular malformations (VMs) are congenital anomalies that involve blood and or lymphatic vessels and originate in fetal life. Vascular malformations grow proportionally with the host and may be unnoticed at birth. The classification of VMs is based on basis according to the type of vessel involved (artery, vein, lymphatic, and combinations) and the morphology (troncular vessels or extra troncular vessels). Venous and lymphovenous malformations are the most common types of vascular malformations while vascular malformations associated with hypertrophy or hypotrophy of the soft tissues and/or bones are typical of syndromic conditions, among which the most popular is Klippel Trenaunay Syndrome (KTS) [1,2]. KTS is a complex condition characterized by the concomitant involvement of soft tissues and vascular vessels (truncular anomalies with aplasia or hypoplasia of deep venous vessels), in which persistence of embryonic veins, soft tissue (fat, muscle) hyperplasia, bone elongation, or shortening and lymphatic vessels involvement (rotation, aplasia, or hypoplasia of the lymphatics) is often observed.

In the last 20 years, in the field of VMs, the application of Next Generation Sequencing (NGS) on tissue specimens has led to a better understanding of the molecular scenario behind these pathologies. Therefore, several recent studies have linked KTS to *PIK3CA* gene somatic variants [3,4], arterio-venous malformations (AVMs) to activating somatic variants in the *KRAS* gene [5,6], and lymphovenous malformations (LVMs) to *TEK* and *PIK3CA* mosaicism [7].

Usually, these molecular approaches are based on surgical biopsies and the biological analysis should be made on fresh tissues. This implies the presence of a genetic laboratory near the surgical ablation site. The two main obstacles underlying biopsies are the risk of bleeding in the presence of arteriovenous shunting and the risk of collecting non-diagnostic samples because of the mosaicism in the distribution of mutated affected cells. Cells involved in vascular anomalies are located principally in vessels or adjacent to them. Thus, we have previously envisioned the possibility of using cfDNA originated from the turnover cells and released in vessels in order to look for causative variants localized in the affected tissues. Since the turnover of these cells could not be as high as in malignant tumors, we thought that the best source of significant amounts of cfDNA was the blood specimen taken as close as possible to the vascular malformation, i.e., the efferent vein. Taking advantage of this approach we have previously shown that indeed *PIK3CA* variants justify a large number of KTS cases [8], while *RAS*-pathway variants generally underlie AVMs [6]. In addition, unexpectedly, we found that somatic mutations in the *MET* gene represent the most relevant alterations in LVMs [9].

In the present work, using this unique non-invasive NGS-liquid biopsy approach, we have molecularly characterized a large cohort of fifty-three patients. We demonstrate that this approach has a very high diagnostic yield with 70% of positive patients and that allows for categorizing VMs, clinically difficult to distinguish, implementing nosological characterization useful for tailored drug therapies.

## 2. Materials and Methods

### 2.1. Patients’ Enrollment and Samples Collection

Fifty-three patients affected by KTS, AVMs, venous malformations, and LVMs underwent genetic counseling and were enrolled at the Medical Genetics Unit of the Azienda Ospedaliera Universitaria Senese, Siena, Italy, for liquid biopsy analysis. Written informed consent for genetic analysis was obtained from all patients. Clinical information was collected in a genetic consultation setting. For six patients, a liquid biopsy from both peripheral blood and efferent veins was performed. In 25 patients, only peripheral blood samples were collected and in 22 patients only lesion efferent vein samples were. The procedure was performed at the Vascular Surgery of Ospedale Maggiore di Crema. All the patients were evaluated by a specialist in vascular malformations and underwent a complete radiological workout including duplex scanning and MRI. Different specimens were archived including formalin-fixed paraffin-embedded (FFPE) tissues for a total of seven patients.

### 2.2. cfDNA Extraction from Plasma

Blood samples (10 mL) were collected from each patient and placed into a cell-free DNA BCT^®^ blood collection tube (Streck, NE, USA). Plasma was stored at −80 °C until cfDNA extraction. cfDNA was extracted from 4 mL of plasma using MagMAX cell-free Total Nucleic Acid Isolation Kit (ThermoFisher Scientific, Waltham, MA, USA), according to the manufacturer’s instructions. cfDNA quality and quantity were verified respectively, using the Agilent™ High Sensitivity DNA Kit (Agilent Technologies, Palo Alto, CA, USA) on Agilent2100 Bioanalyzer (Agilent Technologies) and Qubit™ dsDNA HS Assay Kits on Qubit 3.0 fluorometer (Invitrogen, Carlsbad, CA, USA).

### 2.3. NGS Sequencing on cfDNA

cfDNA libraries were prepared using the Oncomine Pan-Cancer Cell-Free Assay (https://www.thermofisher.com/order/catalog/product/A37664, accessed on 20 June 2022), in which, each sample was tagged with a unique barcode. Four libraries were loaded on an Ion PI kit using Ion Chef (ThermoFisher Scientific, Waltham, MA, USA). Two ion PI chips were sequenced simultaneously using the Ion Proton sequencer (Life Technologies, Carlsbad, CA, USA). The sequence alignment was performed on the human genome assembly GRCh37 (hg19) using the mapping alignment program (TMAP) with default analysis parameters. The variant calling was performed using Torrent Suite version 5.10.1 (ThermoFisher Scientific). For variant annotation, Ion Reporter Software 5.10 (ThermoFisher Scientific) was used. This technology is able to identify various types of alterations, including single nucleotide variants, insertions/deletions, gene fusions, and copy number variations in cancer-related genes (clinical actionable mutations) with a reportable range up to 0.1%. Default setting provides several cutoffs for various parameters; in particular, the molecular coverage of at least 3 with a minimum detection cutoff frequency of 0.065% must be satisfied for the variant calling of SNV/indel. For CNV call the MAPD metric, which is a measure of read coverage noise detected among all amplicons, was set <0.4, the *p*-value < 10^−5^, and the CNV ratio for a copy number gain >1.15-fold change. For sensitive detection of variants with variant allele frequency (VAF) up to 0.1% by variant caller, optimal results are obtained when targeting a median read coverage > 25,000 and a median molecular coverage >2500.

### 2.4. Hematoxylin and Eosin Staining

Different specimens were taken from the upper right limb for patients 19 and 22, from the thorax for patient 18, from the right lower limb and a from lymph node lung cancer for patient 1, from the left lower limb for patient 40, and from the upper limb for patient 22. They were formalin-fixed and embedded in paraffin and then serially cut (10 μm) and rehydrated through 100% xylene and 100, 95, and 70% ethanol before immersion in H_2_O. Sections were then stained with hematoxylin and eosin and dehydrated.

### 2.5. Genomic DNA Extraction from Tissues

For seven patients, gDNA was extracted from FFPE using MagCore Genomic DNA FFPE One-Step Kit for MagCore System (Diatech Pharmacogenetics s.r.l., Ancona, Italy) following the manufacturer’s instructions. gDNA was quantified by Qubit Fluorometer with Qubit dsDNA HS Assay (Life Technologies, Carlsbad, CA, USA).

### 2.6. Genomic DNA Extraction from Blood and Sequencing

Genomic DNA was extracted from EDTA peripheral blood samples using MagCore HF16 (Diatech Lab-Line, Jesi, Ancona, Italy) according to the manufacturer’s instructions. DNA quantity was evaluated using the Qubit 3.0 Fluorometer (Thermo Fisher Scientific, Waltham, MA, USA). Blood DNA was screened for variants using an amplicon-based panel covering all the coding sequences of 17 vascular malformation-related genes (Appendix A). The library was prepared using the Ion AmpliSeq custom panel IAD184982_197 (Life Technologies, Carlsbad, CA, USA) and sequenced on the Ion Torrent S5, according to the manufacturer’s instructions (Thermo Fisher Scientific, Waltham, MA, USA). The sequencing system enables >92% of bases to cover ≥20×. Variants were annotated using Software Torrent Suite v5.0.2 (Life Technologies). Using specific parameters, we removed the adaptors’ contamination and low-quality sequences, so the total amount of clean data was mapped to the UCSC/hg19 reference genome. Indel and variant calls were made using version 2.7 of GATK (Broad Institute, Cambridge, MA, USA). The Software Torrent Suite combines information from variant-based annotation databases (ExAC, 1000-genomes, and avSNP) and the bioinformatics algorithms SIFT, CADD, MutationAssessor, and PhyloP for predicting variant pathogenicity. Detected variants were validated by Sanger sequencing on ABI Prism 330 genetic analyzer (PE Applied Biosystems) and data were analyzed with Sequencher software V.4.9 (Gene Codes, Ann Arbor, MI, USA).

For patients 1, 7, 8, 12, 25, 26, 31, 39, 42, 46, we also performed Exome Sequencing (ES).

For sample preparation, Nextera Flex for Enrichment Panel kit and TruSight One Expanded panel were used. Sequencing was performed on the Illumina NovaSeq6000 System (Illumina San Diego, CA, USA) according to the NovaSeq6000 System Guide. Reads were aligned to the hg19 reference genome using the Burrow–Wheeler aligner (BWA), while variant calling was obtained using an in-house pipeline which takes advantage of the GATK Best Practices workflow. Sequencing with at least a mean coverage >50× and single variants coverage >50× were considered. If single variants coverage was <50×, validation was performed using Sanger sequencing on ABI Prism 330 genetic analyzer (PE Applied Biosystems).

## 3. Results

A total of fifty-three patients affected by different vascular malformations were eligible for NGS-liquid biopsy. During the genetic counseling, the clinical features were registered for all patients and were clinically classified according to the ISSVA classification (Table 1 and Appendix A).

We also performed a liquid biopsy on more than 300 individuals comprising healthy subjects and patients carefully evaluated for different conditions and who did not present vascular malformations. The variants listed in Table 1 were not detected in controls, not even with a VAF below the threshold level. The mean age at the time of cfDNA analysis was 34 years, (range 6 months to 73 years) with 53% (28 out of 53) female patients and 47% (25 out of 53) male patients. Out of fifty-three patients, 37 (70%) reached a molecular characterization, and 16 (30%) were not diagnosed. Six patients underwent liquid biopsy from both peripheral vein and efferent vein. Interestingly in almost all patients, the cfDNA total levels were higher in the efferent vein withdrawal and, in four of them, the variant allele frequencies (VAF), i.e., the percentage of sequence reads matching a specific DNA variant divided by the overall coverage at that locus, were higher when sampling was carried out at the malformation site (Table 2).

Out of 37 positive characterized patients, 12 were clinically classified as KTS patients with ten having the malformation in the lower limbs and two in the upper limbs (Figure 1). Eleven individuals had AVMs, ten had venous malformations and four had lymphovenous malformations. Out of the 16 individuals molecularly negative, three had KTS, four had AVMs, eight had venous malformations and one had a capillary malformation.

In the cohort of KTS patients, we found a recurrent somatic variant in *PIK3CA* c.1633G > A (*p*.(Glu545Lys)) in 25% of individuals (3/12) with a VAF ranging from 0.18%–1.23%. Other *PIK3CA* identified somatic variants are reported in Table 3. For four KTS patients, NGS data from genomic DNA (gDNA) were also available. Interestingly, in patients 8 and 12 germline variants in *COL*-family genes were detected, namely an intronic maternally inherited heterozygous variant in *COL4A4* gene (c.2545 + 143T > A), in patient 8, an eleven-year-old girl, and three heterozygous variants, one pathogenic and two missense (c.1762G > T (*p*.(Glu588*)), c.2686G > T (*p*.(Ala896Ser)) and c.3538C > T, (*p*.(Arg1180cys)) in *COL6A5* in patient 12, a 56-years old male. Patient 1, a 64 years-old female, harbored a heterozygous variant in *HGF* gene c.836G > A (*p*.(Arg279His)).

All AVM-affected patients harbored a variant in a RAS-pathway gene, namely nine out of 11 (81%) AVM-affected patients (median age of 41 years old) (Figure 2) harbored somatic variants in *KRAS* while two out of eleven patients (18%) harbored somatic variants in *MAP2K1*. In seven out of nine individuals the variant in *KRAS* affected codon 12 and was either the c.35G > A (*p*.(Gly12Asp)) (patients 30, 32, 34, 35) or the c.35G > T (*p*.(Gly12Val)) (patients 28, 31, 33) (Table 3). In two patients, 29 and 36, the *KRAS* variant was located in a different codon, codon 61, c.183A > C (*p*.(Gln61His)). Interestingly in these cases, it was associated with a second variant, an *FGFR3* CNV with a VAF of 1.5% in patient 29, and a *MAP2K1* variant c.171G > T (*p*.(Lys57Asn)) with a VAF of 1.59% in patient 36. Two patients (31 and 32), with a somatic variant in KRAS, c.35G > A (*p*.(Gly12Asp)) and c.35G > T (*p*.(Gly12Val)) respectively, additionally harbored germline heterozygous variants in the genomic DNA, i.e., an intronic variant in *KDR* (c.2373 + 10T > A) and a pathogenic variant in *GLMN* gene c.108C > A (*p*.(Cys36*)).

Among the ten patients with venous malformations (median age of 35 years old) (Figure 3), we did not find any recurrent somatic mutation in a specific gene. However, in most of them, a cumulative effect of germline variants in genes implied in vascular angiogenesis and somatic variants in VM-associated genes was observed. Indeed, in this cohort of individuals, we performed NGS analysis on genomic DNA in seven patients, and in four of them, NGS on DNA from tissue specimens was also carried on. Interestingly, in patient 20, who harbored a somatic variant in *PTEN* (VAF = 0.11%) detected by peripheral blood liquid biopsy, we also found a maternally inherited heterozygous genomic variant in *SMAD4* c.1301A > C (*p*.(Tyr434Ser)) and an additional tissue somatic variant in *COL3A1* c.4096C > T (*p*.(Gln1366*)) with a VAF of 0.2%. Efferent vein liquid biopsy in patient 19 and peripheral vein liquid biopsy in patient 22 detected, in the first, a combination of a *CHEK2* variant c.1117A > G (*p*.(Lys373Glu)) with a VAF of 0.25% and an *FGFR3* CNV with a score of 1.43 and, in the latter, a variant in *APC* c.4216C > T (*p*.(Gln1406*)) with a VAF of 0.23%. Interestingly, in both patients 19 and 22, a heterozygous genomic variant in *GLMN*, namely the c.108C > Ap.(Cys36*)) and the c.395-1G > C, was found with no additional variants in the same gene or in additional genes on tissue specimens. In patient 22, the *GLMN* variant was inherited from a healthy mother. Patient 25 who presented a somatic variant in *NRAS*, c.182A > G (*p*.(Gln61Arg)) with a VAF of 0.1%, also harbored a heterozygous germline variant in *GJC2* c.1234C > T (*p*.(Hys412Tyr)) and was a compound heterozygous for a *TNXB* pathogenic variant c.2461C > T (*p*.(Arg821*)) and a missense variant of uncertain significance (VUS) c.2467C > G (*p*.(Leu823Val)). In patient 26, harboring two maternally inherited germline variants in *KRIT1* c.707C > T (*p*.(Ser236Leu)) and *FLT4* c.2860C > T (*p*.(Pro954Ser)), peripheral vein-liquid biopsy identified a somatic *AKT2* variant c.1294_1295delTCinsCG (*p*.(Ser432Arg)) with a VAF of 0.16%.

In the latest series of four patients affected by lymphovenous malformations (Figure 4), NGS-liquid biopsy analysis detected *MET* and *FGFR3* somatic variants in three out of four individuals. In two of them (patients 13 and 14) *FGFR3* and *MET* variants were mutually exclusive, while in two of them (patients 15 and 16) were simultaneously present. The same c.3029C > T (*p*.(Thr1010Ile)) MET variant was detected in cfDNA of both patients 13 and 16 with a VAF of 0.32% and 0.97%, respectively. NGS-tissue analysis, available for patient 15, confirmed mosaicism for the *FGFR3* variant at the malformation site with a molecular frequency of 0.59%. The distribution of all identified genomic alterations is shown in Table 1.

## 4. Discussion

Notably, in the present work, we demonstrate how patients, suffering from various vascular malformations and for whom clinical diagnosis based on anatomopathological signs is often very difficult, can be categorized into much more precise clinical categories according to the identification of even low-grade causative somatic variants. Although a VAF lower than 2% was detected for some variants, the occurrence of either variant in the same gene or of the same variants in patients affected by a similar vascular phenotype and the absence of these variants in a control population further supports their causative role. In KTS patients, cfDNA analysis revealed pathogenic variants in the PIK3CA gene in most of them (9 out of 15 KTS). In the AVMs cohort, we found activating somatic mutations in *KRAS* and *MAP2K1* genes. Interestingly variants in codon 12 in *KRAS* seem to be detrimental and are sufficient alone to cause pathological overgrowth and/or vessels sprouting. According to our data instead, variants in codon 61 could be hypomorphic and would need an additional hit for the malformation to occur (patients 29 and 36). Variants in *MET*, in *FGFR3,* or in both of them seem to explain most of the lymphovenous malformations (4 out of 4 patients). HGF is the only known ligand of the c-MET tyrosine kinase receptor. The binding of HGF to c-MET triggers the activation of the c-MET receptor through autophosphorylation of the cytoplasmic domain and the concomitant recruitment of adapter proteins that enhance cell signaling [10]. Aberration of the HGF/c-MET system is well documented in several human cancers [11], such as non-small cell lung cancer (NSCLC), squamous cell carcinoma of the head and neck (HNSCC) cancer, as well as other carcinomas. The *MET* variants identified in our study are not reported as pathogenic for oncogenesis but could trigger vascular overgrowth with an alternative mechanism.

Intriguingly, we have not found any recurrent mutation in venous malformation patients, but in most of them, we observed a compound effect of germline and somatic variants. NGS on both peripheral blood and lesional tissue was performed on patient 20, presenting with angiodysplastic lesions on visceral sites, cavernous hemangiomas, and intestinal polyposis unmasked both a germline heterozygous VUS on *SMAD4* c.1301A > C (*p*.(Tyr434Ser)), inherited from the healthy mother and a somatic pathogenic variant on *COL3A1* c.4096C > T (*p*.(Gln1366*)) with a VAF of 2%. Germline variants in *COL3A1* are associated with vascular Ehlers-Danlos and vessel aneurysms [12], while pathogenic variants in *SMAD4* cause hereditary hemorrhagic telangiectasia (HHT), an autosomal dominant condition characterized by visceral arteriovenous malformation with intestinal, lung, and central nervous system localization. Thus, we cannot exclude the combined effect of this variant, along with the variant in the *COL3A1* gene, in determining the patient’s phenotype with a double-hit mechanism. These findings highlight that in some patients the vascular phenotype could be due to a compound effect of germline and somatic variants. Indeed, *GLMN* variants have been previously associated with glomuvenous malformations through a double-hit mechanism according to which a second somatic event, in the *GLMN* gene, needs to add up to the first germline variant at the malformation site [13]. According to our data germline heterozygous variants in *GLMN* are not only causative of glomuvenous malformations but could be related to both AVMs and venous malformations according to the second somatic hit. We found the same *GLMN* variant c.108C > A (*p*.(Cys36*)), both in a patient with an AVM and in a patient with venous malformations in association with a somatic *KRAS* variant c.35G > A (*p*.(Gly12Asp)) and a somatic *FGFR3* CNV, respectively.

Notably, according to these data, tailored pharmacological treatment can be proposed on the basis of the molecular nosological classification. Alpelisib has indeed been proven effective for the treatment of PROS (PIK3CA-related overgrowth syndromes) [14] since it acts on *PIK3CA* activating variants. Therefore, pharmacological treatment can consistently help the classic debulking and vascular surgery in molecularly ascertained KTS cases. Activating mutations in the *KRAS* gene lead to the overactivation of the MAPK–ERK pathway in endothelial cells [15]. Using MAP-ERK kinase inhibitors like Mekinist may thus represent a valid approach for the treatment of AVMs. Lymphovenous malformations with *MET* variants could benefit from a pharmacological approach using drugs acting on the HGF/MET pathway as Capmatinib. In conclusion, liquid biopsy, allowing a novel nosological classification, represents an essential tool for defining a more appropriate therapeutic strategy. 

## Figures and Tables

**Figure 1 jcm-11-03740-f001:**
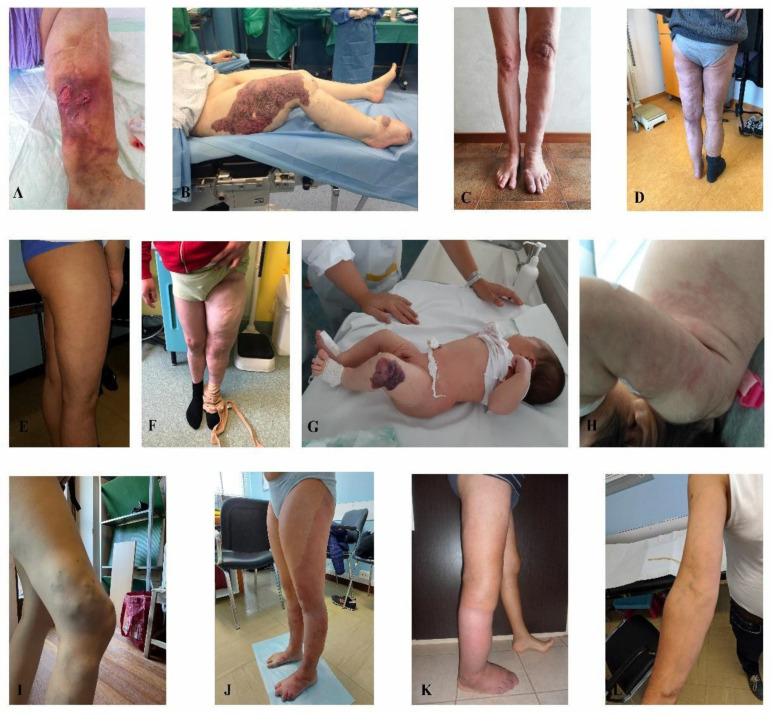
Klippel-Trenaunay syndrome. (**A**) patient 1, (**B**) patient 2, (**C**) patient 3, (**D**) patient 4, (**E**) patient 5, (**F**) patient 6, (**G**) patient 7, (**H**) patient 8, (**I**) patient 9, (**J**) patient 10, (**K**) patient 11, (**L**) patient 12.

**Figure 2 jcm-11-03740-f002:**
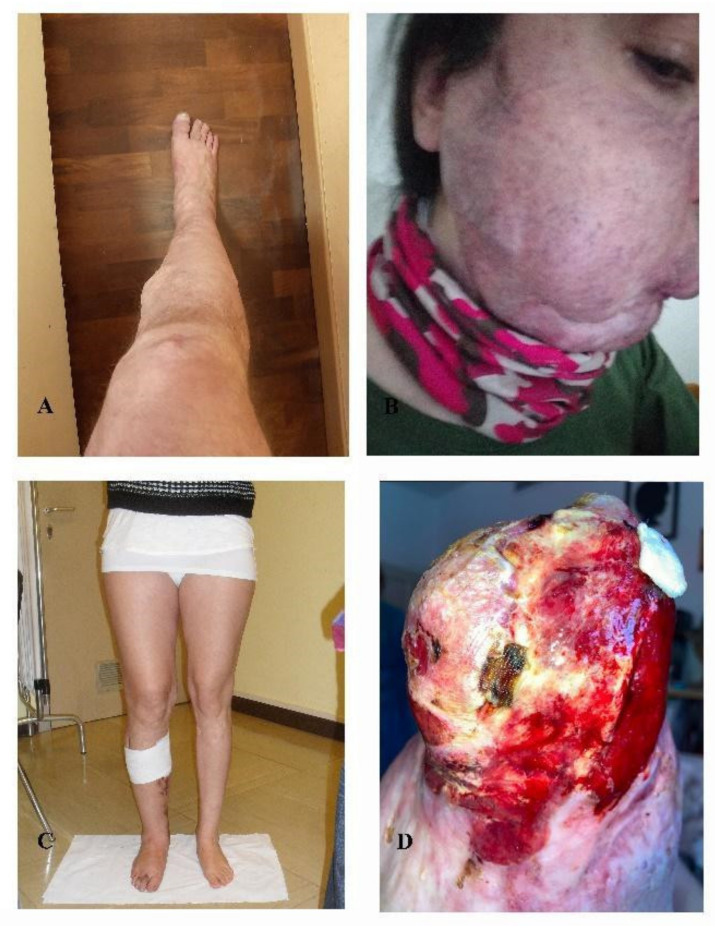
Arteriovenous malformations. (**A**) patient 28, (**B**) patient 29, (**C**) patient 31, (**D**) patient 32. Photos not available: patients 27, 30, 33, 34, 35, 36, 37.

**Figure 3 jcm-11-03740-f003:**
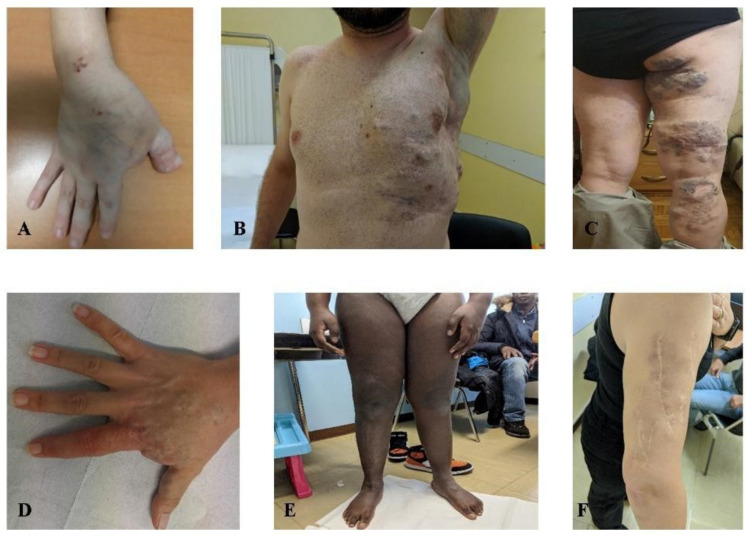
Venous malformations. (**A**) patient 17, (**B**) patient 18, (**C**) patient 21, (**D**) patient 22, (**E**) patient 23, (**F**) patient 26. Photos not available: patients 19, 20, 24, 25.

**Figure 4 jcm-11-03740-f004:**
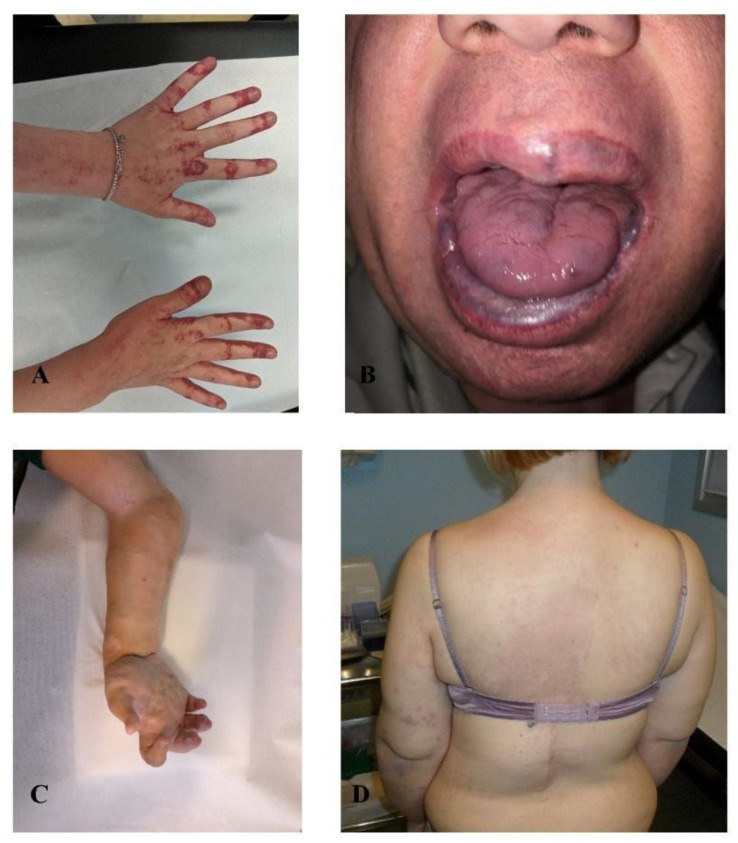
Lymphovenous malformations. (**A**) patient 13, (**B**) patient 14, (**C**) patient 15, (**D**) patient 16.

**Table 1 jcm-11-03740-t001:** Patients’ molecular characterization and clinical classification according to the ISSVA guidelines. The table reports the patient’s identification number, sex, age, mutations identified by peripheral NGS- liquid biopsy and their AFs, mutations identified by efferent vein NGS- liquid biopsy and their AFs, mutations found in gDNA and in tissues specimens by NGS analysis and their AFs.

Patient	Sex	Age	Phenotype	Peripheral NGS Liquid Biopsy, AF (%)	Efferent Vein NGS Liquid Biopsy, AF (%)	gDNA NGS Not Available (N/A) Not Informative (N/I)	Tissue NGS, AF (%)
1	F	64	KTS	*PIK3CA* (*p*.(Glu545Gly)) 0.23%, *TP53* (*p*.(Gly154Val)) 0.21%	*PIK3CA* (*p*.(Glu545Gly)) 0.0783%	*HGF* (*p*.(Arg279His)) heterozygous	*PIK3CA* (*p*.(Glu545Gly)) 1.33%, *PIK3CA* (*p*.(Glu545Gly)) 0.10%
2	F	34	KTS	*PIK3CA* (*p*.(Glu542Lys)) 0.21%	*PIK3CA* (*p*.(Glu542Lys)) 0.75%	N/A	N/A
3	F	58	KTS	N/A	*PIK3CA* (*p*.(Glu726Lys)) 1.09%	N/A	N/A
4	M	38	KTS	*PIK3CA* (*p*.(Glu545Gly)) 1.08%	*PIK3CA* (*p*.(Glu545Lys)) 0.18%	N/A	N/A
5	F	26	KTS	*PIK3CA* (*p*.(Met1043Ile)) 1.47%	N/A	N/A	N/A
6	M	38	KTS	*PIK3CA* (*p*.(Glu545Lys)) 0.96%	*PIK3CA* (*p*.(Glu545Lys)) 1.23%	N/A	N/A
7	M	6d	KTS	N/A	*FGFR3* CNV 1.5, *PIK3CA* (*p*.(Glu545Lys)) 0.32%	*FOXF1* (*p*.(Val378Met)) heterozygous	N/A
8	F	11mo	KTS	*PIK3CA* (*p*.(Glu453Lys)) 0.87%	N/A	*COL4A4* c.2545 + 143T > A heterozygous	N/A
9	F	23	KTS	N/A	*PIK3CA p*.(His1047Arg)) 0.10%	N/A	N/A
10	F	18	KTS	N/A	*FGFR3* (*p*.(Phe384Leu)) 49.9%	N/A	N/A
11	M	64	KTS	*PIK3CA* (*p*.(Glu453Lys)) 0.22%	N/A	N/A	N/A
12	M	56	KTS	*PTEN* (*p*.(Val343Glu)) 0.14%	N/A	*COL6A5* (*p*.(Glu588*)) heterozygous, (*p*.(Ala896Ser)) heterozygous, (*p*.(Arg1180cys)) heterozygous	N/A
13	F	26	Lymphovenous malformation	*MET* (*p*.(Thr1010Ile)) 0.32%	N/A	N/A	N/A
14	M	42	Lymphovenous malformation	N/A	*FGFR3* (*p*.(Phe384Leu)) 2,38%	Negative to MAV genes panel	N/A
15	F	37	Lymphovenous malformation	N/A	*MET* (*p*.(Asp1028Asn)) 0.11%, c.3082 + 2T > C 0.11%, *FGFR3* (*p*.(Phe384Leu)) 0.16%	N/A	*FGFR3* (*p*.(Phe384Leu)) 0.59%
16	F	34	Lymphovenous malformation		*MET* (*p*.(Thr1010Ile)) 0.97%, *FGFR3* (*p*.(Phe384Leu)) 0.68%	Negative to MAV genes panel	N/A
17	F	25	Venous malformation	*PIK3CA* (*p*.(Glu542Lys)) 0.18%	N/A	Negative to MAV genes panel	N/A
18	M	33	Venous malformation	Negative	*MET* *p*.(Thr1010Ile) 1.20%, *FGFR3* (*p*.(Phe384Leu)) 1.34%	N/A	Failed (N/I)
19	F	42	Glomuvenous malformation	N/A	*CHECK2* (*p*.(Lys373Glu)) 0.25% *FGFR3* CNV 1.43	*GLMN* (*p*.(Cys36*)).	Failed (N/I)
20	F	40	Venous malformation	*PTEN* (*p*.(Phe341Leu)) 0.11%	N/A	*SMAD4* (*p*.(Tyr434Ser)) heterozygous inherited from mother VUS	*SMAD4* (*p*.(Tyr434Ser)) heterozygous *COL3A1* (*p*.(Gln1366*))
21	M	37	Venous malformation	N/A	*PIK3CA* (*p*.(Gly12Asp)) 0.30%, *TP53* (*p*.(Arg248Trp)) 0.070%, (*p*.(Val173Met)) 0.076%, (*p*.(Pro152Leu)) 0.09%	N/A	N/A
22	F	49	Venous malformation	*APC* (*p*.(Gln1406*)) 0.23%	N/A	*GLMN* c.395-1G > C heterozygous inherited from mother	Tissue not confirmed LB (N/I)
23	M	14	Venous malformation	*TP53* (*p*.(Pro191fs)) 0.10%	N/A	negative to MAV genes panel	N/A
24	M	18	Venous malformation	*FGFR3* CNV 1.5	N/A	N/A	N/A
25	M	41	Ghouram Stout-Venous malformation	*NRAS* (*p*.(Gln61Arg)) 0.1%	N/A	MAV genes panel *GJC2* (*p*.(Hys412Tyr)), *TNXB* (*p*.(Arg821*)), (*p*.(Leu823Val))	N/A
26	F	37	Glomovenous malformation	*AKT2* (*p*.(Ser432Thr)) 0.16%	N/A	*KRIT1* (*p*.(Ser236Leu)) heterozygous, *FLT4* (*p*.(Pro954Ser)) heterozygous, both inherited from mother, *OPA1* (*p*.(Arg272*)) heterozygous	N/A
27	M	51	AVM	*FGFR2* (*p*.(Ser252Leu)) 1.38%	N/A	N/A	N/A
28	M	35	AVM	*KRAS* (*p*.(Gly12Val)) 0.84%	N/A	N/A	N/A
29	M	45	AVM	N/A	*KRAS* (*p*.(Gln61His)) 6,64% *FGFR3* CNV 1.5	N/A	N/A
30	M	56	AVM	*KRAS* (*p*.(Gly12Asp)) 0.19%	*KRAS* (*p*.(Gly12Asp)) 1.63%	N/A	N/A
31	F	34	AVM	N/A	*KRAS* (*p*.(Gly12Val)) 4,11%	*KDR* c.2373 + 10 T > A heterozygous	N/A
32	M	29	AVM	N/A	*KRAS* (*p*.(Gly12Asp)) 1.18%	*GLMN* (*p*.(Cys36*)) heterozygous	N/A
33	F	45	AVM	N/A	*KRAS* (*p*.(Gly12Val)) 4,19%	N/A	N/A
34	F	40	AVM	N/A	*KRAS* (*p*.(Gly12Asp)) 1.77%	N/A	N/A
35	M	47	AVM	*KRAS* (*p*.(Gly12Asp)) c.35G > A 1.28%	N/A	N/A	N/A
36	M	46	AVM	N/A	*MAP2K1* (*p*.(Lys57Asn)) 1.59% *KRAS* (*p*.(Gln61His)) 0.12%	N/A	N/A
37	F	24	AVM	*MAP2K1* (*p*.(Lys57Asn)) 0.89%	N/A	N/A	N/A
38	M	7 Mo	Capillary malformation	N/A	Negative	Negative to MAV genes panel	N/A
39	F	5	KTS	Negative	N/A	*SLC2A1* (*p*.(Pro3Ser))	N/A
40	F	26	KTS	Negative	N/A	*TGFB2* (*p*.(Val207Leu))	Negative
41	F	61	KTS	N/A	negative	N/A	N/A
42	M	27	Venous malformation	Negative	N/A	*PIEZO 1* (*p*.Arg2491_Glu2492dup)) heterozygous inherited from father, *PIEZO1* (*p*.(Thr1732Met)) heterozygous inherited from mother, heterozygous variant, *TGFRB1* (*p*.(Pro83Leu)) inherited from mother	N/A
43	M	40	Ghouram stout	Negative	N/A	N/A	N/A
44	M	60	venous Malformation	Negative	N/A	N/A	N/A
45	F	30	venous malformation	N/A	Negative	Negative to Rendu-Osler-Weber genes panel	N/A
46	F	40	Bean syndrome	Negative	N/A	*MLH1* (*p*.(Phe626Leu)), *FANCA* c.2778 + 10C > T	N/A
47	M	18	Venous Malformation	Negative	N/A	N/A	N/A
48	F	17	Venous malformation	Negative	N/A	N/A	N/A
49	F	32	Venous Malformation	N/A	Negative	N/A	N/A
50	F	2	AVM	Negative	N/A	N/A	N/A
51	M	73	AVM	N/A	Negative	N/A	N/A
52	F	40	AVM	N/A	Negative	N/A	N/A
53	M	42	AVM	N/A	Negative	N/A	N/A

**Table 2 jcm-11-03740-t002:** cfDNA levels from plasma of both peripheral and efferent vein withdrawals. cfDNA concentrations are reported in the table for each patient. cfDNA total levels were higher in the efferent vein blood withdrawals than in peripheral blood.

Patient’s Code	Peripheral Blood (ng cfDNA)	Efferent Vein (ng cfDNA)
1	46.2	129.6
2	69	27
4	10.5	59.4
6	81.9	111.6
18	3.36	7.71
30	29.58	201.5

**Table 3 jcm-11-03740-t003:** Most recurrent somatic variants in the different types of AVMs and their frequencies in our cohort of patients. Variant type, and its frequency in our cohort, is indicated in each column.

Type of Vascular Malformation	Variant	% Mutation Recurrence
KTS	*PIK3CA* c.1633G > A (*p*.(Glu545Lys))	25% (3/12)
KTS	*PIK3CA* c.1357G > A (*p*.(Glu453Lys))	17% (2/12)
KTS	*PIK3CA* c.1634A > G (*p*.(Glu545Gly))	17% (2/12))
KTS	*PIK3CA* c.1624G > A (*p*.(Glu542Lys))	8% (1/12)
KTS	*PIK3CA* c.2176G > A (*p*.(Glu726Lys))	8% (1/12)
KTS	*PIK3CA* c.3129G > C (*p*.(Met1043Ile))	8% (1/12)
KTS	*PIK3CA* c.3140A > G (*p*.(Hys1047Arg))	8% (1/12))
AVM	*KRAS* c.35G > A (*p*.(Gly12Asp))	36% (4/11)
AVM	*KRAS* c.35G > T (*p*.(Gly12Val))	27% (3/11)
AVM	KRAS c.183A > C (*p*.(Gln61Hys))	9% (1/11)
AVM	*MAP2K1* c.171G > T (*p*.(Lys57Asn))	18% (2/11)
Lymphovenous Malformation	*MET* c.3029C > T (*p*.(Thr1010Ile))	75% (3/4)

## Data Availability

The datasets used and/or analyzed during the current study are available for the corresponding author on reasonable request.

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
