# Peer review of "Nosological and Theranostic Approach to Vascular Malformation through cfDNA NGS Liquid Biopsy"

_jcm, 2022, doi:10.3390/jcm11133740_

Round 1

Reviewer 1 Report

In “Nosological and theranostic approach to vascular malformation through cfDNA NGS liquid biopsy” by Serio etal, the authors isolated cell-free DNA was purified from either peripheral blood (PB) or efferent vein (EV) from 53 patients affected with Klippel Trenaunay Syndrome (KTS), arteriovenous malformations (AVM), venous malformations and lymphovenous malformations (LVMs) diseases and performed Ion Proton sequencing.   

Major Issues

No disease-free control samples were obtained and not every patient was sampled for both PB or EV tissue.  There were no replicate samples obtained from each patient.  Only 7 people were sequenced as gDNA, so how were somatic mutations detected for the other 46 patients.  The goal was to identify circulating DNA variants that could serve as biomarkers for these diseases.

There is no explanation of the coding in Table 1.  For example, what do these values mean for patient 1: “PIK3CA (p.(Glu545Gly)) 0,23%, TP53”  0,23% of what?  What does TP53 mean?  Was it mutated?  I don’t understand the data in this table at all.

I don’t understand the sequence analysis.  The methods state that “Sequencing analysis was performed using Ion Reporter Server System.”  What does this mean?  Did the sequences pass QC?  What was done exactly? It looks like it is only integrating a subset of genes (17?).  Please explain which genes were tested.  Further, there are no details in these words for the gDNA sequencing: “Variants were annotated using Software Torrent Suite, which combines informations from variant-based annotation databases (ExAC, 1000-genomes, and avSNP) and the bioinformatics algorithms SIFT, CADD, MutationAssessor, and PhyloP for predicting variant pathogenicity.”  Again did the samples pass QC?

What does Variant Allele Frequency mean?  Is this relative to a control population like 1000 genomes?  If so, what ancestry corrections were made in these determinations?

In the abstract, the authors state that PIK3CA variants were found in 67% of KTS individuals; KRAS variants in 60% of AVM; MET was mutated in 75% of LVMs.   I have no idea if the experiments actually detected these variants since there was little explanation of the bioinformatics.  I also don’t know if these were cfDNA variants relative to general population (again what genetic background corrections were performed?) or somatic mutations in the patient which seems impossible since the gDNA was only sequenced for 7/53 patients.

Basically, there were no statistics done to show if any of these “markers” are significantly enriched in this cohort.  Maybe it is an issue with communication and lack of details, but all of these data could be noise.

Minor Issues

Page 3:  Typo “anyall patients”

What does “Out of the sixteen molecularly negative individuals” mean on page 7?  These people had no molecules?

What is the purpose of Table 2?  Was the amount of DNA sequenced variable between patients?

Reviewer 2 Report

Dear Authors

It is a great study with enormous prospect, I do conagratulate. But I wish you to make further clarification on a few confusing descriptions as following:

1. Page 2, 2. Materials and Methods. Patients’ enrollment and samples collection: How much is this ‘LVM’ different from KTS? KTS is ‘one of combined form of LM and VM’. Verify it.

2. Page 2, 2. Materials and Methods. Patients’ enrollment and samples collection: What is FFPE tissues?

3. Page 10, Para 1: Figure 4 was designated to lymphovenous malformations but actual patients listed in Fig 4 represent lymphatic malformation with no clear evidence for the venous malformation combined. Explain.

4. Page 11, 4. Discussion, Sentence 2: It is hard to call ‘9 out of 15 KTS’ as ‘almost all’! I suggest other fair description like ‘its absolute majority’, if you would.

All the best,

Reviewer 3 Report

Serio et al reported very interesting data showing the somatic mutation detection in vascular malformations through cfDNA-NGS. They found series of causative mutations in several key genes such as PIK3CA, KRAS, MET, FGFR3, MET in AVM, KTS, and VM, etc. These data are consistent with the reported NGS results from biopsy tissues in the field. This is an excellent study to show the merits of cfDNA-NGS liquid biopsy for vascular malformation diagnosis, potentially providing a molecular guidance for a tailored treatment based on patient's mutation classification.   

I have couple minor comments for authors consideration.

(1) One limitation is lack of health control subjects. Many of samples showed a mutation frequency lower than 1% in table 1; What are the base line mutation frequencies of those causative mutations in cfDNA in healthy population?

(2) KTS patients H & J in Figure 1 also presented port wine birthmarks on the skin. What is the GNAQ(R183Q) mutation status in the cfDNA from both patients?

(3) the exact pathogenic role of low frequency somatic mutations (less than 2%) is yet to be determined. Such a limitation should be addressed in the discuss section.

(4) In the method section, the authors stated that "detected variants were validated by Sanger sequencing..."; but there is no data showing in the manuscript. I would suggest to remove this sentence to avoid the confusion that Sanger seq is generally unable to detect a mutation frequency lower than 10% due to the fluorescent background noise.  

Author Response

Please see my attachment.

Round 2

Reviewer 1 Report

The authors have responded to my request for more detailed Methods.  They argue against my concern that the variants they discovered are biomarkers by saying they are presenting the method.  I am OK with this.

Author Response

Thanks for the suggestion. According with the reviewer we have adjusted the captions of the tables and slightly modified some sentences in english.

Hoping that results more clear,

best regards